# Predicting Student Well-Being: Network Analysis Based on PISA 2018

**DOI:** 10.3390/ijerph17114014

**Published:** 2020-06-05

**Authors:** Elena Govorova, Isabel Benítez, José Muñiz

**Affiliations:** 1Department of Psychology, University of Oviedo, 33003 Oviedo, Spain; elena@estudiosyevaluaciones2e.com (E.G.); jmuniz@uniovi.es (J.M.); 2Department of Methodology of Behavioral Sciences, University of Granada, 18071 Granada, Spain

**Keywords:** well-being, bullying, network analysis, PISA 2018, teaching style

## Abstract

The latest trends in research extend the focus of school effectiveness beyond students’ acquisition of knowledge and skills, looking at aspects such as well-being in the academic context. Although the concept of well-being itself has been defined and measured in various ways, neither its dimensions nor the relationships between the components have been clearly described. The aim of the present study was to analyse how the elements of well-being interact and determine how they are influenced by school factors. To do that, we conducted a network analysis based on data from the Programme for International Student Assessment (PISA) 2018 international assessment. Our results demonstrated that cognitive, psychological, and social well-being variables form a solid welfare construct in the educational context, where students’ resilience and fear of failure, along with their sense of belonging, play central roles. Although the influence of school factors on student well-being is generally low, teaching enthusiasm and support promote positive school climates which are, in turn, crucial in reducing bullying.

## 1. Introduction

One of the main priorities of the United Nations Educational, Scientific, and Cultural Organization’s (UNESCO) strategy on education is the promotion of children and young people’s well-being [1]. The Convention on the Rights of the Child established four general principles that should be uniformly and universally adhered to: non-discrimination; the best interests of the child; the right to life, survival, and development; and respect for the views of the child [2]. Despite that, statistics in the educational arena indicate that these principles are a long way from being reality. In primary education, the data are extremely worrying. According to the International Association for the Evaluation of Educational Achievement (IEA) assessment, 29% of 4th grade students reported that they suffered bullying each month and 14% reported being bullied on a weekly basis [3]. In addition, the 2018 Programme for International Student Assessment (PISA) reported that the average percentage of frequently bullied students reached 8% in secondary education across countries members of the Organisation for Economic Co-operation and Development (OECD). Furthermore, 9% of 15-year-old students reported always being sad and around 6% reported being miserable and scared. More than 30% of students felt that their teachers did not understand them and did not listen to their points of view, and only about two thirds of students reported that they were satisfied with their lives [4]. These data suggest that there is still work to be done in terms of improving children’s rights and that there are numerous factors influencing children’s experiences. In this regard, well-being in the educational context, the school environment, and experiences in the school context seem to be important in understanding the situation and encouraging changes that improve students’ lives. Ultimately, “children and young people spend so much of their childhoods in this context” [5]. 

Students’ well-being is an objective that the school must set as a priority, since it is increasingly considered as an indicator of the quality of the teaching-learning process. The well-being concept is complex, as it englobes a variety of elements like physical and mental health, the happiness and satisfaction of students’ lives, and socialization and interaction with peers and teachers, among others. The present research aims to assess how these elements are related to each other and which of them play a central role in school-related quality of life. We hope that our work contributes to a better understanding of the role that students’ well-being plays in the school context. This will allow us, on the one hand, to design actions and projects to improve the effectiveness of educational systems and, on the other, to improve the overall quality of life of students, which will ultimately lead to an improvement in their education.

### 1.1. Well-Being in the Educational Context

The OECD defines well-being as a dynamic state characterized by students experiencing the ability and opportunity to fulfil their personal and social goals [6]. The promotion of student well-being is already an important item on the agenda in education. Taking that idea, Chapman [7] points to four reasons for justifying the inclusion of well-being as a relevant variable. Those reasons are based on the importance of well-being for health, educational achievement, socialization and social values, and the development and formation of the human mind.

What happens in school is key to understanding whether students enjoy good physical and mental health, how happy and satisfied they are with different aspects of their lives, how connected to others they feel, and the aspirations they hold for their future [8,9,10]. For instance, a positive class atmosphere where effort is encouraged and rewarded and in which children are accepted and supported by their teachers regardless of their intellect and temperament can have a positive effect on student well-being [11,12]. As Slee and Skrzypiec [13] describe, interventions in school for enhancing student well-being are focused on improving relationships, resilience, and the school climate and reducing bullying. However, the connection between these variables and the reported well-being is not clear.

Nor is the concept of well-being clear. Despite many studies working on defining well-being, it is not static. Currently, there are many theoretical approaches differing in the elements that make up well-being and the key focus of the construct, as summarized by McLellan and Steward [14]. There are also several studies collecting qualitative data about students’ perceptions of well-being. For instance, in a project in New South Wales students defined well-being through feelings such as happiness or the absence of sadness, harmonious social relationships, and being a moral actor in relation to oneself or behaving well towards others [15]. Another qualitative study in Australia indicated that self-esteem, self-respect, and self-confidence were central to student well-being [5].

In the international context, definitions of well-being try to cover the full concept by including elements common to students from different countries and cultures and usually by including indicators related to students´ school-lives which depend on their personal and family lives. For instance, the definition formulated by the Organisation for Economic Co-operation and Development (OECD) indicates that student well-being refers to the psychological, cognitive, material, social, and physical functioning and capabilities that students need to live a happy and fulfilling life [16]. 

Based on that definition, in the 2015 edition of PISA the framework for the analysis of student well-being described the construct according to five domains [6]: (1) cognitive well-being, which includes variables related to student knowledge and abilities for resolving everyday issues; (2) psychological well-being, which includes perceptions of the students about their own lives, their engagement with school, and their plans for the future; (3) physical well-being, which refers to students’ health and their habits related to sports and eating; (4) social well-being, which evaluates how students perceive their relationships within and outside of school; and (5) material well-being, which refers to the available resources for meeting students’ needs.

However, the analysis of the student responses to the well-being scales in PISA 2015 demonstrated that the proposed structure of the well-being concept could not be confirmed, as some of the domains were not unidimensional and some of the indicators included in the domains were not relevant to student welfare [17]. Therefore, the subsequent edition of PISA in 2018 incorporated some changes to the definition of well-being. Based on the same framework, the indicators and the composition of the domains were adjusted in order to better reflect variables defining student well-being. In the present study, the model of well-being is based on both PISA editions.

Figure 1 illustrates the proposed well-being model which is configured according to data collected in PISA 2018. The model evaluated in the present research includes three well-being dimensions: the social dimension as defined in PISA 2015 and the psychological and cognitive dimensions as defined in PISA 2018. In the psychological dimension, students’ life satisfaction, sense of meaning in life, and feelings are used as indexes for measuring subjective well-being [18]. These variables evaluate the general perception of life satisfaction and sense of meaning in life as well as emotions and moods. Self-efficacy and fear of failure inform about students’ perceptions of their own general abilities to deal with challenging circumstances [19,20]. The cognitive dimension focuses on the growth mindset, which evaluates the extent to which someone perceives their abilities and intelligence as variables that can be developed [21]. Finally, the social dimension reflects students’ perceptions about the level of cooperation and competitiveness at schools, as well as social acceptance and levels of bullying.

Reports produced using PISA 2018 and 2015 data provide important information about how these variables of student well-being behave and how the values for each of them relate to performance. However, in order to more deeply understand the concept of well-being in students, we would need to know how the domains and indicators from each domain relate to the others. In addition, knowing how these components connect to variables related to the school and teachers’ activities would help to formulate policies focused on increasing students’ well-being.

### 1.2. Network Analysis in Education

From a methodological perspective, the structure of well-being and other psychological variables can be studied by following various approaches. For instance, exploratory and confirmatory factorial analyses and other clustering methods provide information about groups of items measuring the same dimension. Furthermore, information about the strength of the relationships between variables can be estimated through correlation or regression indices. Nevertheless, these procedures cannot illustrate connectivity in such a way that changes in the structure indicate meaningful differences.

Networks have recently been gaining importance as models that represent complex phenomena in human behavioural science [22], such as psychopathology [23,24] and personality [25,26,27]. In the educational field, for example, network analysis has been applied to learn about the influence of multiculturality on student motivation [28] and understand the role of interest in science in student involvement [29].

The aim of the present study was to describe the concept of well-being in the educational context. To do that, we first analysed the elements making up the concept of well-being itself and the nature of the interactions between those elements. Following that, we assessed the relationships between well-being dimensions and school factors, such as school climate and teaching practices. After proposing a model illustrating relationships between well-being and school factors, we compared the network models representing the relationships between well-being and school factors between groups.

## 2. Methods

### 2.1. Sample

We based our analysis on the data collected in PISA 2018, the latest edition of the international student assessment that aims to provide information about the ability of 15-year-old students to face the challenges in their future lives. PISA measures students’ capacity to use their knowledge and skills in reading, mathematics, and science and to apply them to real life situations. The OECD organizes PISA assessments every three years. In PISA 2018, 37 OECD countries and 47 associated countries and economies participated in the study. The advantage of PISA data is that, along with performance measurement, it offers a variety of contextual data which enables in-depth studies on the different students’ factors, including well-being. In recent years, the OECD has prioritized not only the research on students’ performance but also the assessment of well-being and social and emotional skills, creating comprehensive frameworks for their definition and measurement.

Table 1 reflects the sample configuration for the OECD and for subgroup samples used in this study. For high and low performing subsample configurations, the students were classified according to their performance in PISA. The average performance of each student in the three domains measured in PISA—reading, mathematics and science—was estimated and the students were classified according to their total performance in the top and bottom quartiles. The students from the highest performance quartile were considered high performers and students from the lowest quartile were considered low performers. This approach is commonly used in the OECD secondary analysis and publications based on PISA data for the measuring of the gap between the highest and lowest-performing students in a specific domain in order to compare the learning outcome parity [30].

Table 2 represents the sample configuration for each country where data was analysed separately (sample size and percentage of girls), along with the country abbreviation used throughout the study. Of the 37 OECD countries that participated in PISA 2018, we selected 26 with data available for the whole range of variables examined. The countries with missing data for any of the variables used in the network analysis were excluded.

### 2.2. Instruments

The PISA 2018 assessment comprised over 15 h of testing in three main domains—reading, mathematics, science—and in a novel domain specific to each PISA (in 2018, global competence). The participating students took different combinations of test items during a 2 h assessment session. In each PISA cycle, one domain is tested in detail, taking up nearly half of the total testing time. The main area of assessment in 2018 was reading, as it was in 2000 and 2009. In most countries, the test was computer-based. In reading, a multi-stage adaptive methodology was applied in computer-based tests. The students were assigned a block of test items based on their performance in previous blocks.

Students and authorities at participating schools (e.g., school principals) also completed context questionnaires. The students’ questionnaire collected information about their background context, attitudes towards learning, habits in and outside of school, opinions about school resources and teaching practices, perceptions of the learning environment, and their motivation and engagement. In PISA 2018, five additional questionnaires were offered as options: a computer familiarity questionnaire, well-being questionnaire, educational career questionnaire, parent questionnaire, and teacher questionnaire. The specific well-being questionnaire covered a wide range of variables aimed at collecting information about students’ life satisfaction, social connections and activities, subjective well-being, and family support; however, only three OECD countries chose this option. In the present study, we used the data from the student questionnaire which was common to all the countries to assess well-being. This questionnaire also included a notable set of variables aimed at measuring this aspect of students’ lives. The whole range of these variables was included in the network analysis. They are listed in Table 3 along with a brief description.

The selection of the variables related to school factors was also determined by their availability in the PISA dataset. Therefore, we considered the whole set of variables that measure teaching practices or teaching methodology. In PISA 2018, the student questionnaire included only one variable that provided information about the school environment from the students’ perspective—disciplinary climate. The list of school level variables is presented in Table 4.

Table 5 reflects the internal consistency of the well-being scales at the OECD level, measured with Cronbach’s alpha, as well as the number of items composing each of the scales. Following the commonly used cut-off criteria (0.9 excellent, 0.8 good, 0.7 acceptable), all scales have shown acceptable values and, therefore, the internal consistency of the scales is confirmed. In the PISA Technical Report corresponding to each PISA edition, the internal consistency coefficients at the country level are also presented [31]. The variables that did not show an acceptable internal reliability were not considered for the scale construction.

PISA uses advanced statistical and psychometric approaches for the estimation of scores of latent traits from multiple observed responses both for performance measures and context latent constructs. In order to ensure the comparability between groups in terms of the invariance of item parameters across groups of participating countries and language groups therein, the same set of estimated item parameters is held in each group surveyed [32]. In 2015, PISA adopted an innovative approach in order to evaluate if equal item parameters can be assumed across countries. With this purpose, international item and person parameters were estimated based on all the participants across all groups, and the difference between the observed item characteristic curve and the model-based item characteristic curve was calculated with the use of the root mean square deviance (RMSD) item-fit statistic. The cut-off criteria of 0.3 was established, with larger values indicating that the international item parameters were not appropriate for this group [31]. Thus, this approach quantified how well the international parameters described the observed data of each country. The item parameter estimation as well as the estimation of latent traits were conducted using Item Response Theory scaling methodology, applying the generalised partial credit model (GPCM) [33].

The original version of the student questionnaire can be found in Annex A of the PISA 2018 Assessment and Analytical Framework [34].

### 2.3. Data Analysis

The network approach allows the representation of interactions between the elements of the phenomena and lets us understand the structures and consequences of these interactions. It allows us to interpret the relationship between different elements simultaneously, the underlying reciprocal influence, and interconnections [24,35]. Network analysis pursues two types of hypotheses: those that seek to understand the cause of the formation of determinant interactions in a given population, and those that aim to discover how the interaction of the elements influences the outcomes [36]. In the present study, the network analysis was based on both premises; first, we studied the relationship between the well-being variables and then we investigated the influence of school factors on these relationships.

The network approach is based on two principal concepts: nodes, which represent the elements of a model, and edges, the connections between nodes that represent their pairwise interactions. Once the network is computed, different tools or indices can be used to summarize the patterns of relationships in the network. The centrality indices of the network allow us to explore the relative influence of a node in the context of other nodes [37]. They mean that we can analyse the relative importance of the node within the network based on the connection pattern [38]. Several measures of centrality can be estimated: betweeness centrality, closeness centrality, and strength centrality. Strength centrality refers to the magnitude of the association of the node with the other nodes and which node has the strongest connections [39]. Closeness centrality indicates which nodes can better predict others. It is defined as the inverse of the sum of the distance from one node to all the other nodes in the network. A node with a high closeness centrality will be affected quickly by changes in any part of the network [37]. Betweeness centrality is defined as the number of times a node is between two other nodes. A node with a high value of betweeness indicates that it is well connected with the rest of the network nodes.

We performed the network analysis using the R-package qgraph [40], both for the overall estimation of well-being variable interactions and for the exploration of the relationships between the school factors and well-being dimensions. We also carried out a network comparison for different subgroups: girls vs. boys and high performers vs. low performers. We conducted pair-wise comparisons of the invariance of the overall network structure and the global connectivity with the Network Comparison Test (NCT) package in R [41]. The comparison of the overall network structure shows if there is a pattern in the unique interactions between the variables in the network of each individual group, and a comparison of the overall connectivity lets us see if the strength of these interactions between variables is similar [42,43].

### 2.4. Procedure

We performed the network analysis in two stages. First, we examined the co-occurrence of well-being variables that form part of each dimension for the whole set of OECD countries. Each variable represented an independent node grouped in three well-being dimensions. The centrality indexes were estimated for each node, measuring the importance and influence of each node over others.

In the second stage, we added the school-level variables to the well-being network: the teaching style and the disciplinary climate at school. This network was also built for the overall OECD sample and for each of 26 countries with available information. Finally, we conducted the network comparison tests for pairs of groups based on gender and performance level (distinguishing between high and low performers).

## 3. Results

### Student Well-Being Network

The first stage of the network analysis was an analysis of the well-being dimensions as defined in PISA. Figure 2 shows the network representation of the relationship between the well-being dimensions at the OECD level. The visual inspection of a network is always a very useful first step, providing important information with minimal effort [44]. Green lines indicate positive connections and red lines negative connections. Thicker lines represent stronger connections and thinner lines represent weaker connections. A visual inspection of the network shows that the theoretical configuration of well-being dimensions is confirmed empirically; the variables making up each dimension are strongly related to each other, either positively or negatively. They are also demonstrated to be related to variables in the other dimensions, which reiterates the idea of the complexity of the well-being concept.

The figure also illustrates the strength of these interactions. In the psychological dimension of well-being, the strongest relationship is between life satisfaction and positive feelings. Life satisfaction is also strongly connected to the perception of meaning in life, which at the same time is positively related to resilience in terms of self-efficacy. Students who believe in their own capacity, especially when facing difficult circumstances, and feel that their lives have meaning and purpose are also generally more satisfied with their lives.

The interaction between learning goals and motivation to master tasks is also strong and positive, and both variables demonstrate a positive relationship with student resilience. A more positive attitude towards competition is related to higher task mastery motivation and is also linked to higher resilience. The fear of failure is negatively associated with life satisfaction and resilience in terms of self-efficacy and is also negatively connected to the sense of belonging, a variable in the social well-being dimension.

In the social well-being dimension, the sense of belonging is a strong predictor of bullying. Students who feel integrated and who do not feel awkward or out of place in their school report less exposure to bullying. On the contrary, higher levels of bullying are related to the perception of more competitive environment in schools. The students who report suffering from bullying indicate that the students in their school seem to value competition, share the feeling that competing with each other is important, and feel as though they are being compared to others.

Lastly, the only cognitive dimension variable, growth mindset, is negatively associated with the fear of failure. Students with a growth mindset reported less fear of failing than students with a fixed mindset.

Figure 3 shows the centrality indexes for the well-being network. The perception of higher self-efficacy, or resilience, is the node with the highest influence over the others. This variable has the highest strength centrality; in other words, it has strong direct connections with many nodes. It also demonstrates the highest value of closeness centrality, as the direct and indirect paths that connect it to other variables are relatively short, which means that it is more vulnerable to changes in the other variables. Fear of failure is another node with high levels of centrality and demonstrates the greatest betweenness centrality, as it lies on the shortest paths between other nodes. In line with the conclusions resulting from Figure 2, life satisfaction and a sense of belonging also have important roles in moderating the well-being network.

In the second stage of data analysis, we added school factors related to teaching style and school climate to the well-being network model in order to assess the potential impact of these factors on well-being (Figure 4). The results show that the connections between the well-being variables and teaching style are mainly weak and insignificant, although important insights can be gleaned.

The teaching style variables, considered as an individual network, form a well-defined set of elements with positive significant interactions between nodes. Higher teacher enthusiasm is strongly connected to more frequent teacher activities that stimulate reading engagement. The teaching style based on individual help and adapting lessons to class needs and knowledge is usually accompanied by students perceiving more feedback about their performance and areas for improvement. Teacher-directed instruction is highly correlated to the perception of teacher support. Through teacher enthusiasm and reading engagement stimulation the teaching style impacts the school environment as measured by the perception of the disciplinary climate. The higher the educator engagement and involvement, the lower the frequency of occasions when there is noise and disorder in language lessons, when students cannot work well, or when they do not pay attention to what the teacher says. At the same time, the students who report positive a disciplinary climate also perceive a less competitive environment along with higher co-operation levels between peers. In the previous stage of data analysis, high levels of competitiveness were shown to be associated with more frequent bullying, while in contrast, fellowship and support between peers reduced bullying exposure rates, leading to the conclusion that teaching style can actually moderate student well-being more directly through a positive disciplinary climate.

Figure 5 shows the centrality indexes for the variables making up the proposed network models. The values are presented for the OECD average, as well as for the population subgroups—female and male students—and high and low performers.

In the enriched network model, along with the variables with high levels of centrality observed in the first stage of data analysis (exposure to bullying, sense of belonging, and resilience), teacher enthusiasm stands out as the teaching style factor with most influence on the rest of the nodes. In other words, teacher enthusiasm improves students’ perceptions of adaptive teaching, individual approach, and cognitive stimulation.

There is practically no variation in the strength of interactions between network elements for the female and male subpopulations, whereas the centrality indexes are less homogeneous for high and low performers. Thus, a sense of belonging is relatively more important for high-performers’ well-being, while motivation to master tasks is more significant for the well-being of low performers.

In order to provide an objective measurement of potential network invariance for gender and performance samples, we carried out a network comparison test. The results (Table 6) showed that the pattern of unique interactions between indicators in the network was not completely identical across subpopulations, as the overall network structure invariance was not supported.

Lastly, in order to check how the interactions between the set of elements in the proposed networks varied between countries, we configured networks for each of 26 countries with available information. Figure 6 shows comparisons of the centrality indexes by country. Each dot represents the centrality indexes for each country. We see that the distribution of strength centrality indexes for each variable are relatively homogeneous between countries, whereas the distribution of betweeness centrality is more variable. This means that while the strength of direct connections between nodes is relatively more similar between countries, the moderating roles of the variables in terms of vulnerability and intermediation do not follow the same patterns in all of them.

## 4. Discussion

The aim of the study was twofold. On the one hand, we sought to explore the overall interactions between the well-being variables assessed in PISA 2018, and on the other, we examined the connections between school climate and teaching style and student welfare. The network approach allowed us to visualize the well-being model as an integrated system and represent the complex interactions between welfare concepts, highlighting the most influential elements.

Student resilience in terms of self-efficacy was a central area in the psychological dimension of the well-being network. In PISA, the resilience or ability of a living being to adapt to a disruptive agent or an adverse situation or state [45] is measured through students’ perceptions about their self-efficacy in general competences. Students who believe in their abilities in spite of adversities are also highly motivated, persistent, and enjoy working hard. These observations are in line with previous studies, highlighting that resilient students are self-confident, autonomous, and demonstrate high levels of self-control [46,47]. They are ambitious and satisfied with their lives, reporting that their life has satisfactory meaning. Among the OECD countries, greater self-efficacy was also associated with higher performance in reading, especially between countries performing below the OECD average [4]. Moreover, resilient students are less likely to be afraid of failure.

Fear of failure is another well-being element with a high level of centrality. It is related to less life satisfaction and a lower growth mindset. The students who do not believe that their effort in learning is a source of their success are more prone to doubt plans for their future or to feel incompetent and untalented when they fail. Previous research has already shown that fear of failure is associated with high levels of worry and somatic anxiety and low levels of optimism, and can have significant impacts, such as the experience of embarrassment, the loss of self-esteem, or being uncertain about the future [20]. It is especially worrying that in the OECD countries, many 15-year-olds expressed a fear of failure. Around 56% of students agreed or strongly agreed that when they fail they worry about what others think about them, and 55% were afraid of not being talented enough when they fail [4].

These data show the importance of promoting student self-esteem, resilience, and self-efficacy, both in general and specific competences, along with the use of strategies that reduce anxiety and the fear of failure. Several student and school factors have been shown to have an impact on resilience; these include ambitious educational aspirations, valuing learning and instrumental motivation, student beliefs about their teachers’ confidence in their abilities [48], and higher reading confidence [49]. Yeager and Dweck [50] stated that, to be resilient, students need to be taught strategies to overcome challenges rather than self-esteem boosting or trait labelling. Some teacher strategies that have been shown to have a positive impact on motivational-affective outcomes include subject domain-specific activities for processing information (e.g., mathematics problem solving, science inquiry), social experiences (e.g., cooperative learning, student discussion), time for learning, and regulation and monitoring (e.g., providing feedback and support, teaching students self-regulation strategies, and monitoring) [51]. At the country level, policies based on grouping students according to their socioeconomic background are negatively associated with the percentage of resilient students, while teacher support plays an essential role in encouraging self-efficacy [52]. At the same time, the fear of failure can be reduced by enhancing a growth mindset in students. Setting challenging goals and providing opportunities for equal competences or content acquisition, along with designing learning environments accordingly, can be a successful teaching strategy [4].

The central element of the social well-being network is the sense of belonging. It is strongly connected to psychological well-being variables, positive thinking, and cooperation, and negatively related to the fear of failure. However, most importantly the greater the perception of a sense of belonging, the lower the frequency of exposure to bullying. When students feel accepted, included, and respected by the community, they are less likely to suffer from bullying. A sense of belonging is also associated with higher academic performance [4,53]. Previous research has shown that supportive student–teacher relationships had a positive impact on the sense of belonging [54], although in the current study we did not see any significant interactions. Nevertheless, the negative influence of a competitive school environment on student engagement was confirmed.

The school-level factors measured in terms of teaching style did not demonstrate significant direct impacts on student well-being. These results are in line with previous studies demonstrating low school effects on student welfare [17,55]. Nevertheless, the network modelling showed the importance of a positive disciplinary climate that improves student–peer cooperation while the perception of competitiveness decreases. These two factors in turn reduce the frequency of bullying at school.

Large-scale assessments show that the promotion of student welfare in the educational context is still pending for most OECD countries. It has been widely agreed that schools’ roles go well beyond the acquisition of cognitive knowledge and competences. Student perceptions of the school environment are key for understanding the source of students’ fears and anxieties and the level of students’ social engagement. Appropriate teaching styles and strategies can contribute to the improvement of this environment. Defining and designing them should be undertaken by the educational community along with academic curriculum development.

Lastly, the comparison of network structures across subpopulations showed that the unique interactions between well-being and school variables differ for female and male students, high and low performers, and at the country level. These results suggest that the design of intervention policies should be based on the particular network models of each subpopulation that will make it possible to precisely locate the main elements that could be strengthened.

The present paper provides evidence on the composition of the well-being concept related to student and school factors. That contribution is relevant, as it allows understanding the relative weight of educational variables on students’ lives. Educational policies and practices will benefit of the results of the present study, as they will have guidelines for working on those dimensions which can help to improve the experiences of students in the school. The model proposed helps to learn about priorities when focusing on students´ well-being beyond their performance.

The main limitation of the study lies in the fact that the configuration of the network model is conditioned by the availability of the reduced set of well-being measures in the PISA 2018 questionnaires. For instance, the cognitive dimension is presented through only one variable, whereas in previous PISA cycles several constructs, such as enjoyment of learning and instrumental motivation, made up a solid cognitive well-being dimension. Moreover, some other important well-being components, such as the physical dimension, were not assessed. Future research will aim to configure an integral and comprehensive well-being model in the educational context and focus on comparing the interactions between well-being indicators between populations of different ages and socioeconomic characteristics.

## 5. Conclusions

This research aims to provide evidence on the complex interactions between well-being components and the school context, applying the novel approach of social network analysis. Well-being networks for OECD countries and an extended well-being–school environment network were constructed. Student resilience and fear of failure were shown to be the most influential elements in the psychological well-being dimension, whereas the sense of belonging played a central role in social well-being. Although teaching style did not seem to directly impact well-being, it affected the disciplinary climate, which in turn moderated competitiveness and cooperation and therefore influenced rates of bullying.

## Figures and Tables

**Figure 1 ijerph-17-04014-f001:**
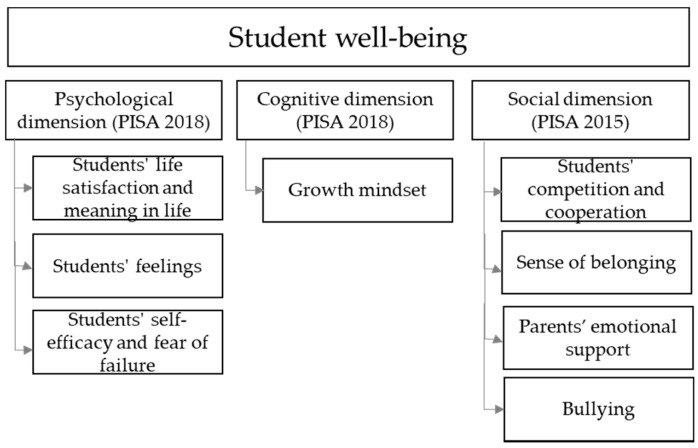
Well-being model based on Programme for International Student Assessment (PISA) 2018 and PISA 2015. This figure has been prepared by the authors and is based on figures presented in the PISA 2018 Results, Volume III (OECD, 2019), and the PISA 2015 well-being framework [6].

**Figure 2 ijerph-17-04014-f002:**
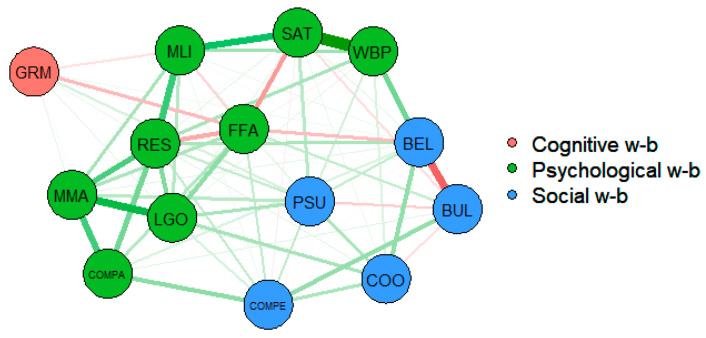
Network representation of the relationship between well-being dimensions, Organisation for Economic Co-operation and Development (OECD) average. GRM = growth mindset; LGO = learning goals; MMA = motivation to master tasks; RES = self-efficacy; FFA = fear of failure; MLI = meaning in life; WBP = positive feelings; SAT = life satisfaction; COM = attitudes towards competition; BUL = exposure to bullying; COMPER = student competition; COOPP = student co-operation; PSU = parents’ emotional support; BEL = sense of belonging.

**Figure 3 ijerph-17-04014-f003:**
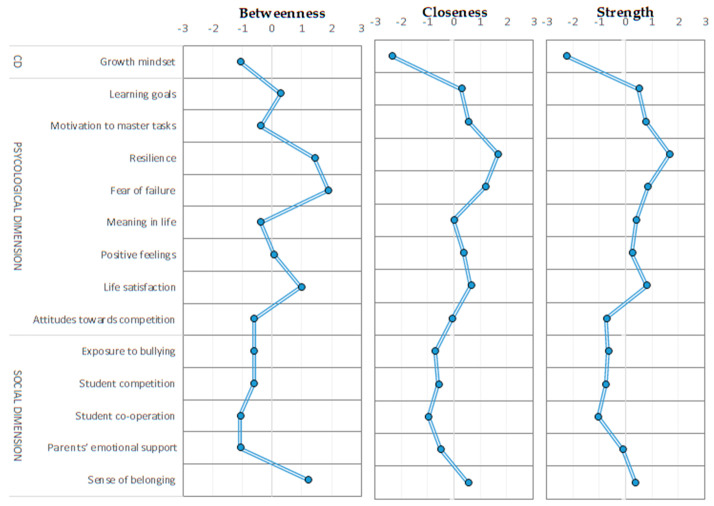
Centrality indexes of well-being dimensions, OECD average. CD = cognitive dimension.

**Figure 4 ijerph-17-04014-f004:**
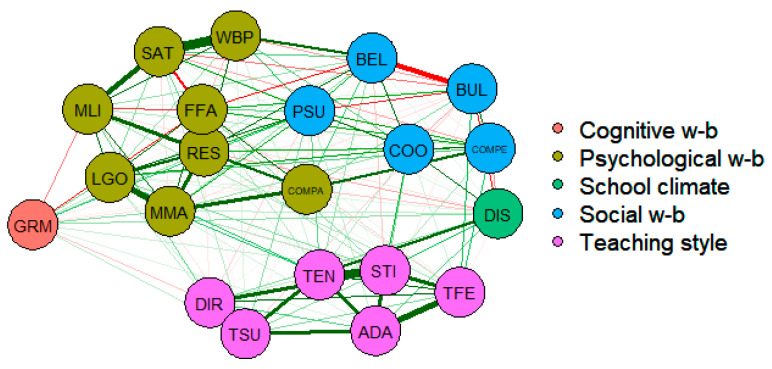
Network representation of the relationship between well-being dimensions, teaching style, and school climate, OECD average. GRM = growth mindset; LGO = learning goals; MMA = motivation to master tasks; RES = self-efficacy; FFA = fear of failure; MLI = meaning in life; WBP = positive feelings; SAT = life satisfaction; COM = attitudes towards competition; BUL = exposure to bullying; COMPER = student competition; COOPPE = student co-operation; PSU = parents’ emotional support; BEL = sense of belonging; ADA = adaptive instruction; TEN = teacher enthusiasm; DIR = teacher-directed instruction; TFE = teacher feedback; STI = teachers’ stimulation of reading engagement; TSU = teacher support; DIS = disciplinary climate.

**Figure 5 ijerph-17-04014-f005:**
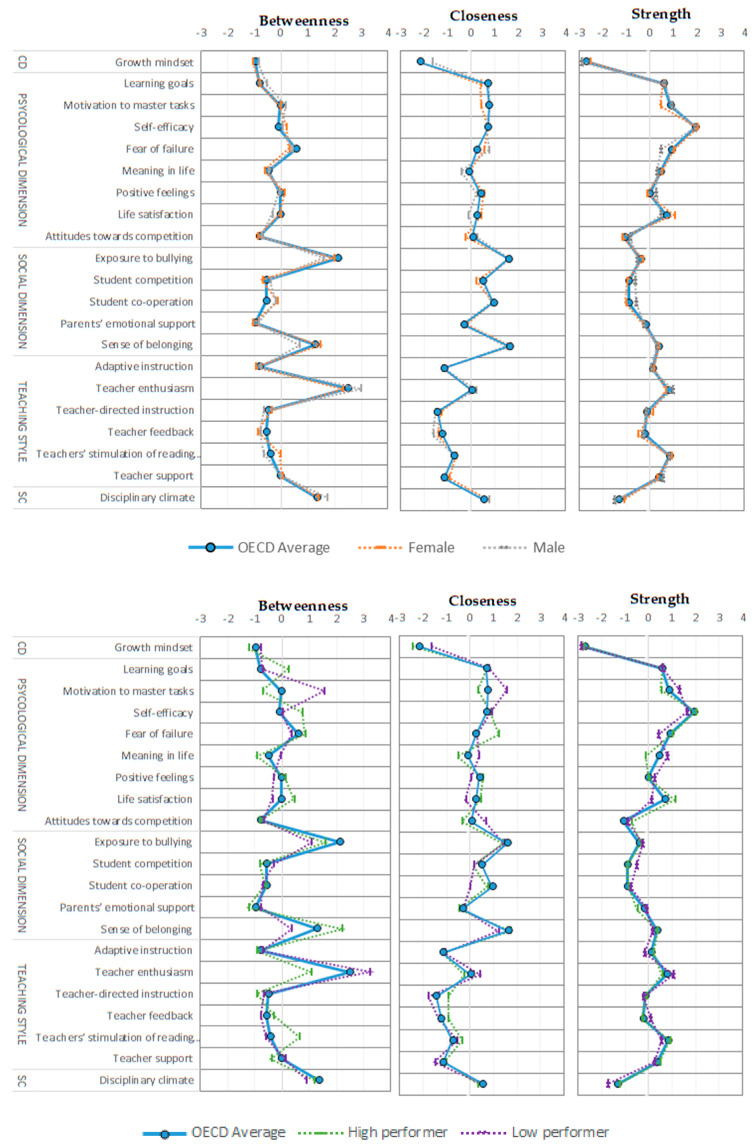
Centrality indexes of well-being dimensions, teaching style and school climate, OECD average, female and male students, and high and low performers. CD = cognitive dimension; SC = school climate.

**Figure 6 ijerph-17-04014-f006:**
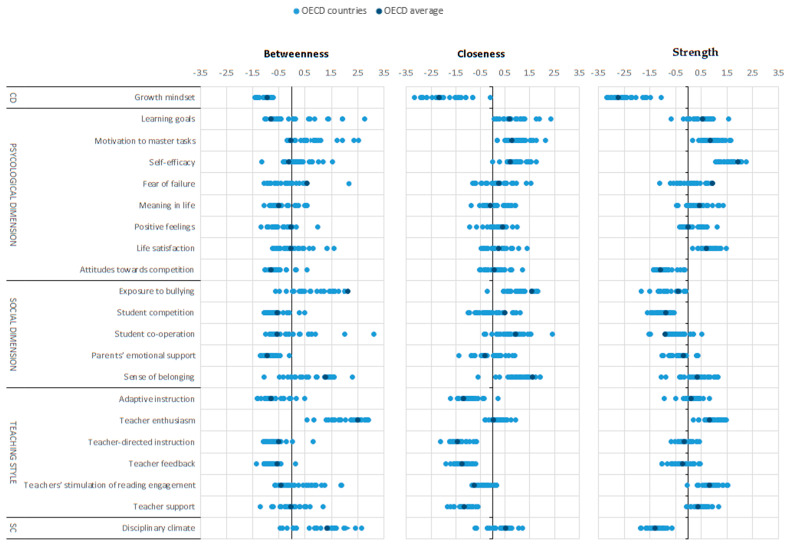
Centrality indexes of well-being dimensions, teaching style, and school climate at country level. CD = cognitive dimension; SC = school climate.

**Table 1 ijerph-17-04014-t001:** International sample configuration.

	Total	Percentage of Total
Total OECD *	294,527	-
Female	146,674	50%
Male	147,851	50%
High performers	73,630	25%
Low performers	73,630	25%

* Organisation for Economic Co-operation and Development.

**Table 2 ijerph-17-04014-t002:** Country-level sample configuration.

Abbreviation	Country	Total	% Girls	Abbreviation	Country	Total	% Girls
AUT	Austria	6802	49%	IRL	Ireland	5577	50%
CHE	Switzerland	5822	48%	ISL	Iceland	3296	50%
CHL	Chile	7621	50%	JPN	Japan	6109	51%
COL	Colombia	7522	51%	LTU	Lithuania	6885	49%
CZE	Czech Republic	7019	50%	LUX	Luxembourg	5230	50%
DEU	Germany	5451	46%	LVA	Latvia	5303	51%
ESP	Spain	35,943	50%	MEX	Mexico	7299	52%
EST	Estonia	5316	50%	NLD	Netherlands	4765	49%
FIN	Finland	5649	49%	POL	Poland	5625	51%
FRA	France	6308	49%	SVK	Slovak Republic	5965	50%
GBR	United Kingdom	13,818	51%	SVN	Slovenia	6401	47%
GRC	Greece	6403	50%	TUR	Turkey	6890	49%
HUN	Hungary	5132	51%	USA	United States	4838	49%

**Table 3 ijerph-17-04014-t003:** Description of well-being indicators.

Dimensions	Variables	Description
**Cognitive dimension**	Growth mindset	This variable reflects the student’s belief that their intelligence is something that can be developed over time.
**Psychological dimension**	Learning goals	This index is intended to reflect the level of students’ ambitions in learning; whether their objective is to learn as much as possible, master the classroom material, and understand the material thoroughly.
	Motivation to master tasks	This index measures whether the student finds satisfaction in working as hard as they can and in improving their performance, if they are persistent at finishing proposed tasks or mastering material they are potentially not good at.
	Resilience (or self-efficacy)	This index reflects the level of student perception about their self-efficacy; whether they are proud of accomplishing things and if they can easily resolve difficult situations, whether they are able to handle several things at the same time and if they believe in themselves.
	Fear of failure	Fear of failure includes the student’s insecurity and lack of confidence in their abilities. Students with a higher fear of failure worry about what others think about them, are afraid that they are not talented enough, and start doubting their plans for the future when they fail.
	Meaning in life	Meaning and purpose in life, or eudaemonia, refer to students’ beliefs that their lives have satisfactory meaning and that they are aware of what brings meaning to their lives.
	Positive feelings	This index reflects the frequency with which the students normally feel happy, joyful, and cheerful.
	Life satisfaction	This variable corresponds to the students’ overall evaluation of their lives. Specifically, the students indicate how satisfied they are with their life as a whole these days on a 0–10 scale.
	Attitudes towards competition	This index reflects student´ competitiveness, how important competition is for the student, and if it improves their performance. Specifically, the students report whether they enjoy working in situations involving competition with others, if it is important for them to perform better than other people on a task, or if they try harder when they are in competition with other people.
**Social dimension**	Exposure to bullying	This index reflects how frequently students experience bullying, considering three of its expressions: being threatened by other students, being left out of things on purpose, and being ridiculed by other students.
	Student competition	This index reflects the environment of competitiveness in schools as perceived by students. Each participant reports if the students in their schools seem to value competition, if they compete with each other, and if they share the feeling that competing is important.
	Student co-operation	This index reflects the environment of co-operation in schools as perceived by students. Each participant reports if the students in their schools seem to value co-operation, if they co-operate with each other, and if they share the feeling that co-operating is important.
	Parents’ emotional support	This index reflects whether students feel that their families support them emotionally. Specifically, whether their families encourage them to be confident, support their educational efforts and achievements, and support them when they are facing difficulties at school.
	Sense of belonging	The index of sense of belonging reflects students’ feelings about the level of integration and social connections: if they make friends easily at school and if they feel that other students seem to like them or if, in contrast, they feel awkward, out of place, or lonely in their school.

**Table 4 ijerph-17-04014-t004:** Description of school factors.

Dimensions	Variables	Description
**Teaching style**	Adaptive instruction	The students perceive adaptive instruction when the teachers adapt the lesson to their class’s needs and knowledge, provide individual help, or change the structure of the lesson on a topic that most students find difficult to understand.
	Teacher enthusiasm	The teacher enthusiasm variable corresponds to students’ perceptions of the level of teachers’ involvement in, motivation for, and enjoyment of their work. They report if it is clear to them that the teacher liked teaching and dealing with the topic of the lesson and if the teacher´s enthusiasm was inspiring.
	Teacher-directed instruction	This variable refers to the teaching practice when the teacher is the one who transmits the knowledge and controls learning processes in the classroom. In PISA 2018, this index includes students’ responses about the frequency with which teachers set clear goals for their learning, ask questions to check whether they have understood what was taught, or are told what they have to learn.
	Teacher feedback	Perceived teacher feedback refers to students’ opinions about the level of feedback they receive on their strengths in the subject, the areas that should be improved, and how they can be improved.
	Teachers’ stimulation of reading engagement	The index of teachers’ stimulation of reading engagement indicates that the teachers frequently encourage students to express their opinion about a text, help them relate the stories they read to their lives, or pose questions that motivates students to participate actively.
	Teacher support	This index summarizes the student’s perceptions about the support they receive from their teachers. Specifically, it reflects whether the teacher shows an interest in every student’s learning, gives extra help when needed, or continues teaching until all the students understand the subject.
**School climate**	Disciplinary climate	The disciplinary climate index reflects order in the classrooms and the adequacy of the learning environment. Higher values of the index indicate that in most lessons students listen to what the teacher says, there is silence and order, there is no need to wait a long time for students to quiet down, and students in general can work well.

**Table 5 ijerph-17-04014-t005:** Internal consistency of the well-being variables and school factors.

Dimensions	Variables	Number of Items	Internal Consistency
Cognitive dimension	Growth mindset	1	-
Psychological dimension	Learning goals	3	0.861
	Motivation to master tasks	4	0.773
	Self-efficacy	5	0.795
	Fear of failure	3	0.815
	Meaning in life	3	0.870
	Positive feelings	3	0.828
	Life satisfaction	1	-
	Attitudes towards competition	3	0.793
Social dimension	Exposure to bullying	6	0.861
	Student competition	4	0.853
	Student co-operation	4	0.915
	Parents’ emotional support	3	0.897
	Sense of belonging	6	0.769
Teaching style	Adaptive instruction	3	0.768
	Teacher enthusiasm	4	0.885
	Teacher-directed instruction	4	0.786
	Teacher feedback	3	0.874
	Teachers’ stimulation of reading engagement	4	0.847
	Teacher support	4	0.883
School climate	Disciplinary climate	5	0.871

**Table 6 ijerph-17-04014-t006:** Results of pair-wise Network Comparison Test (NCT) for network structure and global connectivity invariance.

	Network Structure	Global Connectivity Invariance
Network for female subpopulation–network for male subpopulation	0.12 **	0.27 **
Network for top performers–network for low performers	0.20 **	0.21 **

Note. Network structure invariance: M-statistic, network global connectivity invariance: S-statistic; ** = significantly different network structure, global connectivity if *p* < 0.05.

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
