# Peer review of "Predicting Student Well-Being: Network Analysis Based on PISA 2018"

_ijerph, 2020, doi:10.3390/ijerph17114014_

Round 1
Reviewer 1 Report
Overall, this paper did a good job achieving it's primary goal - to understand the connection between different measures of well-being, and how these factors are associated with school influences. The analyses were advanced, which was good, but there were a number of limitations preventing this paper from reaching it's full potential. See below, in order as they appear in the manuscript:
- At the first mention of an abbreviated entity (i.e., PISA, UNESCO), spell out what the entity is.
- The introduction was missing citations for some of their claims.
- When well-being is first mentioned, it should be defined.
- The new dimensions are confusing based on their wording – there are three listed in the table, but in the first paragraph that they are mentioned, only two are listed. This should be clarified.
- Aren’t there other factors associated with the cognitive dimension, such as their ability to critically think, learn, or process information? Where did they authors choose these specific characteristics to describe these three dimensions? What support do they have? This is an exceptionally weak area of the paper, yet it has the most importance.
- How were high and low performers defined? As a combination of all three subjects? It’s unclear who these students are. Have other studies used this approach?
- It isn’t clear why the PISA data was selected for this study. What advantage does this dataset have compared to others?
- For their measures of well-being by dimension – how many items per factor? Was internal consistency acceptable? How do the authors know what they are capturing is valid? This issue is also prevalent with school factors. More information about their measures are needed in order to justify using this data.
- How does their analytic approach address the goal of the study?
- An issue throughout the paper is not emphasizing the importance of the study. Why is this study important? What are this study’s contributions and implications?
- There were minor grammatical issues throughout the manuscript.
Author Response
Dear reviewer,
First of all, we would like to thank you for a thorough review and extremely valuable comments. We made changes in the manuscript in order to implement all your suggestions. Hereunder we respond to each of the queries and we describe how we formulated the changes in the paper:
At the first mention of an abbreviated entity (i.e., PISA, UNESCO), spell out what the entity is.
The abbreviated entities have been spelled out when first mentioned in the text (lines 23 and 33).
The introduction was missing citations for some of their claims.
We included some new references in the introduction section in order to provide support for the ideas and statements explained.
When well-being is first mentioned, it should be defined.
A definition of well-being has been included at the beginning of the first section (lines 43 and 44).
The new dimensions are confusing based on their wording – there are three listed in the table, but in the first paragraph that they are mentioned, only two are listed. This should be clarified.
We made changes in the paragraph introducing Figure 1 in order to clarify how the model used is composed by dimensions from two different editions of PISA. In the current version, the three dimensions in the figure are fully described (lines 87-98).
Aren’t there other factors associated with the cognitive dimension, such as their ability to critically think, learn, or process information? Where did they authors choose these specific characteristics to describe these three dimensions? What support do they have? This is an exceptionally weak area of the paper, yet it has the most importance.
Thank you for raising that relevant point. We completely agree with the reviewer about the need of considering additional variables as part of the dimensions. However, in the present study we use the definition of well-being as it is measured in the PISA study. In different editions of PISA, different variables have been assessed as part of the concept of well-being. The current definition reflects the variables considered from the OECD as relevant for measuring well-being, which are the available variables in the dataset.
The additional variables indicated by the reviewer such as the ability to critically think or to process information are also included in the PISA data collection but they are not part of the well-being definition, which is the focus of the present study. In any case, we highlight that fact as a primer limitation of the study.
Nevertheless, there is an additional set of extremely important variables related to cognitive well-being that is not used in the present study, like instrumental motivation, the enjoyment of learning, the perception of self-efficacy in cognitive tasks or the interest in reading or broad subjects. Unfortunately, the selection of the variables that form part of each well-being area is conditioned by their availability in PISA questioners and this set of variables has been modified in the latest PISA edition, 2018, removing most of cognitive well-being variables. One of the future steps of our research group includes the development of the comprehensive measurement instrument of well-being, in each of its dimensions, with particular interest on cognitive and physical constructs, and its consequent validity.
How were high and low performers defined? As a combination of all three subjects? It’s unclear who these students are. Have other studies used this approach?
In page 4 we added additional information for explaining how high and low performing students are grouped (lines 143-150). The approach that we use in this study is commonly used in the OECD secondary analysis and publications based on PISA data for the measuring of the gap between highest- and lowest-performing students in a specific domain in order to compare learning outcomes parity (OECD, PISA-based Test for Schools, 2017).
It isn’t clear why the PISA data was selected for this study. What advantage does this dataset have compared to others?
PISA study, developed by the OECD, is an international assessment that not only evaluates the performance of 15-year students in different competences, but also offers a broad catalogue of contextual data, both from students and schools. That wide dataset enables in-depth studies about students and school factors, including well-being with responses of more than 600.000 students from 79 countries. PISA offers an open access to the whole range of collected data which gives a great opportunity for investigating relevant variables for students using a representative and international sample. In recent years, the OECD is prioritizing not only the research in students’ performance but also the assessment of the well-being and social and emotional skills, creating comprehensive frameworks for their definition and measurement. Summarizing, PISA dataset provides very rich information in different fields of educational context which is available for each participating country and that is the reason why we decided using PISA data. We have also highlighted the advantages of PISA data in the paper (lines 132-141).
For their measures of well-being by dimension – how many items per factor? Was internal consistency acceptable? How do the authors know what they are capturing is valid? This issue is also prevalent with school factors. More information about their measures are needed in order to justify using this data.
We included a new table (table 5 in the current version) where data about internal consistency of scales is described. In addition, we added two new paragraphs (before and after the table) for explaining how the psychometric properties and equivalence between groups are confirmed in PISA studies (lines 191-214).
How does their analytic approach address the goal of the study?
The goal of the study was to describe the concept of well-being in the educational context in terms of the elements making up the concept of well-being itself and the nature of the interactions between those elements. The network analysis approach has been selected as the most appropriate for this goal as it permits to represent the interactions between the elements of the complex phenomena and to understand structures and the consequences of these interactions. The contribution of the analytic approach to reach the goal is emphasized in the first paragraph of the discussion section.
An issue throughout the paper is not emphasizing the importance of the study. Why is this study important? What are this study’s contributions and implications?
A new paragraph has been added in the discussion section (lines 456-461) where the contributions of the study are described as well as the implication of the results for educational policies and practices.
There were minor grammatical issues throughout the manuscript.
The paper was reviewed by a Native American speaker with wide experience in academic papers review. We have conducted an additional thorough review of grammar and style making some minor modifications.
Reviewer 2 Report
This paper tackles an important subject, the development and use of quality-of-life measures in assessing students, with the aim of providing information for educators world-wide (especially in the OECD). There are two interested communities of readers - social scientists and educators. The first are interested in the highly technical aspects of variable development and coordination, the second are interested in knowing what evidence supports which kind of changes in educational practice to address the challenges to students identified at the beginning. I would suggest more clear separation of the interests of these two groups - the social scientists need to appreciate the (to me) quite complicated refinement of the variables, while the educators need to know what is known now about how to make education better. Two other points: readers would like to see more about the validity of the students' responses - how do students feel about revealing personal data on these questionaires? And also, what country-by-country variability has been found and how has it been evaluated?
Author Response
Dear reviewer,
First of all, we would like to thank you for a thorough review and extremely valuable comments and suggestions. We completely agree that the paper has two potential groups of readers, each of them with specific interests. In order to better respond to these interests we have applied the following modification:
We have incorporated more extensive information about the assessment instruments including the validity statistics for variables and constructs used in the study, the explanation of the country-by-country validity process, the analytical approach for internal consistency and item invariance measurement, as well as additional references for methodological documents that can be consulted by social scientists’ interests in well-being assessment (lines 191-214). Unfortunately, PISA assessment does not collect information regarding students´ opinion or how students feel about revealing personal data on these questionnaires. However, the PISA study defines a complex protocol for assuring that students data keeps anonymous and they apply security actions for saving the data. In addition, the instructions provided to schools and students participating in PISA promote the students participation in order to reach a high response rate. In addition, we implemented some changes focused on increasing the technical information for social scientists but also the details about practical implications for educators and policy makers.
Round 2
Reviewer 1 Report
I appreciate the authors responding to my initial concerns with the paper. I feel that the paper has greatly improved. I think the importance of the study could be emphasized in the first few paragraphs of the paper, and the decimal spacing in Table 5 is inconsistent (some have 3 numbers after the decimal; others have fewer). Addressing these minor issues will help the paper reach its fullest potential.
Author Response
Dear reviewer,
Thank you again for your time, your work, and your effort in improving our paper. We added a new paragraph as part of the introduction section where we emphasized the importance of the study as well as our expected contributions to the educational field. We also adjusted the table 5 to unify the presentation of the decimals.
Reviewer 2 Report
Excellent paper!
Author Response
Dear reviewer,
Thank you again for your time, your work, and your effort in improving our paper.